# Augmented ultrasonography with implanted CMOS electronic motes

Yihan Zhang 1,2, Prashant Muthuraman[1], Victoria Andino-Pavlovsky[1], Ilke Uguz[1], Jeffrey Elloian 1 & Kenneth L. Shepard 1✉

Modern clinical practice benefits significantly from imaging technologies and much effort is directed toward making this imaging more informative through the addition of contrast agents or reporters. Here, we report the design of a battery-less integrated circuit mote acting as an electronic reporter during medical ultrasound imaging. When implanted within the field-of-view of a brightness-mode (B-mode) ultrasound imager, this mote transmits information from its location through backscattered acoustic energy which is captured within the ultrasound image itself. We prototype and characterize the operation of such motes in vitro and in vivo. Performing with a static power consumption of less than 57 pW, the motes operate at duty cycles for receiving acoustic energy as low as 50 ppm. Motes within the same field-of-view during imaging have demonstrated signal-to-noise ratios of more than 19.1 dB at depths of up to 40 mm in lossy phantom. Physiological information acquired through such motes, which is beyond what is measurable with endogenous ultrasound backscatter and which is biogeographically located within an image, has the potential to provide an augmented ultrasonography.

[1] Department of Electrical Engineering, Columbia University, New York, NY 10027, USA. [2] School of Integrated Circuits, Peking University, Beijing, P. R. China. ✉email: shepard@ee.columbia.edu

There have been decades of technological advances in the field of ultrasound medical diagnostic imaging. With pioneering work[1] dated back to the 1950s, medical ultrasound, or ultrasonography, has evolved to become a routine imaging procedure for real-time tomography. Compared to other medical tomographic techniques, it has many advantages because of the unique properties of ultrasound waves. In addition to being nonionizing, the low acoustic energy loss in tissue (~0.5–1 dB/cm/MHz[2,3]) allows for significant imaging depth while high spatial resolution is possible because of the short wavelength characteristic of acoustic energy in tissue (about 0.5 mm at a typical carrier frequency of 3 MHz). Imaging frame rates are typically in excess of 25 frames per second (fps) due to advances in imaging hardware and software.

At the same time and motivated by many of the same properties that make it attractive for diagnostic imaging, ultrasound is emerging as a means for powering and communicating with implanted medical devices[4–8]. Wireless powering is attractive because it eliminates the need for batteries, which introduce extra safety concerns and consume considerable volume. What is particularly advantageous for ultrasound in these applications is that attenuation in tissue is significantly less for ultrasound when compared with electromagnetic waves at comparable wavelengths[9]. Wavelength is important because it determines minimum antenna sizes and, consequently, the size of the implanted devices themselves. The low absorption of ultrasound in tissue also supports average power densities as high as 7.2 mW/mm² without deleterious effects[10]. For electromagnetic radiation, the specific absorption ratio (SAR) limits incident average power densities to less than 0.22 mW at 1.6 GHz, for example, in the case where an antenna is positioned at the tissue surface[11].

To achieve a µW-level power budget, previous devices based on ultrasound for wireless powering rely on focused, almost continuous ultrasound "beams". Traditional digital modulation techniques can be then implemented on top of such beams for data transmission, like on-off keying (OOK)[12], and pulse-width modulated amplitude-shift keying (PWM-ASK)[6,13]. Uplink data can be captured in the acoustic domain once the device is properly powered, in the form of either backscattered incident ultrasound[5,14,15] or active transmission of ultrasound energy from the implant[12]. In all of these implementations, accurate knowledge of the implant location is required for power delivery and communication. This location information can be obtained either through an additional imaging session prior to the operation of the implant or a designed localization process on top of normal operation. One such designed process uses a sharp ultrasound pulse emitted from the implant to back-calculate the delays in each transmitter element for focusing, but the implant needs to harvest some power before the chirp can be emitted[16–18]. Another example extracts this delay information from higher-order harmonics of the implant's reflected acoustic waves[19]; but without an active signature, harmonics in the tissue, long used in ultrasound harmonic imaging[20], can potentially shadow this information. Even if the implant can be localized, in many in vivo settings, continuous movement due to muscle activity, heartbeat, and respiration can perturb device location, requiring frequent recharacterization to maintain focus.

In this work, we instead integrate the operation of our custom-designed ultrasound mote[21] with medical sonography, allowing them to be biogeographically located in the image, to be powered by the ultrasound imager, and to communicate back through the image. The mote stays continuously powered when located within the ultrasound imaging sensor's field-of-view (FoV). Bi-directional data transmission is established synchronously with the imaging frame rate and the uplink information (from implant to the imager) can be retrieved during image reconstruction.

Multiple motes can be deployed within a given FoV with non-interfering parallel operation, as digital uplink data signatures are spatially separated in the reconstructed image. These digital data signatures give the motes contrast relative to the surrounding, continuously moving biological environment and can be used to localize them at the resolution of the resulting B-mode image. These sensors allow for an augmented ultrasonography, delivering real-time, biogeographically related physiological information from multiple locations on top of traditional ultrasound tomographic imaging.

The use of imaging-mode ultrasound imposes a restricted power budget on the motes, typically nano-Watts (nW) or below, as well as a data rate limited to the imaging frame rate, which is typically in the range of 10 s of Hz. Fortunately, these data rates are sufficient for most physiological parameters, such as temperature, blood pressure, pH, and most biomarkers. There are many CMOS sensor systems available that operate with nW or sub-nW power consumption that would be compatible with this power envelope at the bandwidths of interest[14,22–24]. Even in the absence of integrated sensing, motes such as these transmitting unique identifiers could find application in real-time tracking of surgical sites, such as in intramedullary nailing[25,26].

## Results and discussion

Figure 1 shows how ultrasonography-compatible motes interact with ultrasound imaging. We consider brightness-mode (B-mode) imaging here, but these devices can operate with any pulse-echo-mode ultrasound imaging technique. When placed deep in tissue and within the FoV of an ultrasound imaging probe, the implant appears due to a local mismatch in acoustic impedance as a bright spot in the reconstructed image. Unlike a passive physiological structure, however, the mote modulates its backscatter from frame-to-frame, making the motes easily identifiable in the ultrasound image (see Supplementary Movies S1 and S2). This modulation also provides the data uplink. The data downlink is, in turn, implemented by changing the width of the ultrasound pulses emitted from the probe during imaging. Instructions and measured physiological data can be sent through

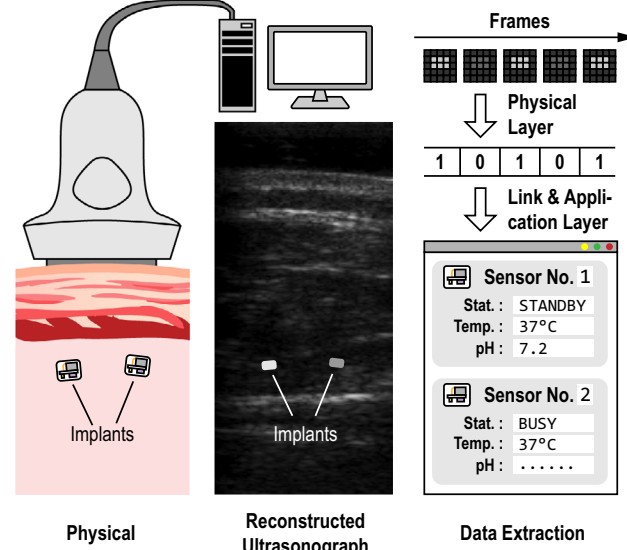

**Fig. 1 CMOS mote implementing augmented ultrasonography.** Illustration of a typical use case; after the motes are physically implanted, they can be identified in an ultrasonograph, and data can be transmitted bidirectionally between the imager and the mote to retrieve real-time physiological information.

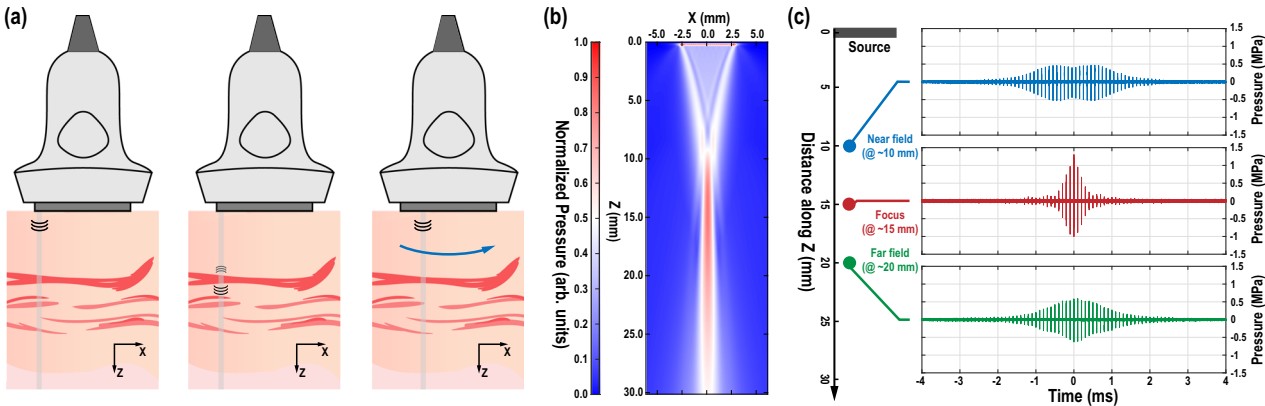

**Fig. 2 Principles of ultrasonography. a** Illustration of the pulse-echo imaging principle; **b** a simulated beam profile using a linear array transducer; and **c** measured pulse envelope at different distances from the source linear array.

this bi-direction data link, with predefined data transmission protocols. Thus, from a communications perspective, these ultrasonography-compatible motes establish a physical layer information channel, upon which data can be delivered reliably for sensing applications at higher protocol layers.

**B-mode sonography and challenges in the physical layer.** Figure 2a shows the operating principle of B-mode sonography that guides the physical layer implementation. Traditional B-mode image frames are generated by scanning a focused ultrasound pulse over the field-of-view and measuring the back-reflected echoes[27]. The depths of all the impedance discontinuities along the path of traversal of the pulse are determined by measuring the delay between the received echoes and the source pulse. Each pulse travels along a line in the axial (or $z$) direction at a certain transverse (or $x$) coordinate, as shown in Fig. 2a, known as a scanline or beam. In a linear array probe, this beamforming is implemented through phased pulsing of multiple elements around the transverse position of interest. Scanning in the transverse direction is implemented by changing this phasing or by changing the elements employed to produce a given beam. In our case, we employ a Verasonics L12-3V linear array probe, a 192-element linear ultrasound transducer array, in which 31 elements at a time are phased to produce a focused, $z$-directed pulse at a given position in $x$ along the transducer array, which is then scanned to produce a frame. To maintain a frame rate of 50 Hz, for example, a total of 192 pulses are emitted at a 100-μs time interval for each frame, and the selection of the 31 transducers is shifted by one in the $x$ direction for each pulse.

Harvesting energy in the context of B-mode sonography must be done from an ultrasound energy that is very sparse in time. For the frame rate of 50 fps employed in our case, each 1-μs-long pulse provides power to the mote with a duty cycle of 50 parts per million (ppm). The energy that is available for power harvesting scales as approximately 0.33 pJ/(kPa)² for each pulse with energy at a MHz-level center frequency. The energy harvesting circuits must be able to respond to these frequencies without significant static power consumption from biasing and leakage currents. Energy storage in the form of decoupling capacitors is required to maintain continuous powering. At a 50-ppm duty cycle and a minimum incident pressure of 400 kPa, which is typical for the imaging studies here, we find that the decoupling capacitance must be at least 100 pF. Making the decoupling capacitance larger than minimum value, however, extends the start-up time, the time interval between when the imaging probe first find the mote to the time the mote generates a distinguishable data signature,

since a supply voltage must reach at least 1.2 V before the mote can begin to operate.

Data transmission from the mote to the imager takes advantage of the way B-mode imaging is performed. Figure 2b shows the simulated ultrasound beam profile with beamforming from 31 transducers focused to a depth of 15 mm, showing the peak pressure detected at each spatial location in the beam. In the near-field region as determined by the transducer and focal depth, the ultrasound energy is more diffuse. Beyond the focal depth in the far-field, the energy distribution pattern becomes narrower but still depends on the distance from the probe. This finite spatial extent of the ultrasound beam means that each mote will both receive and backscatter many pulses in each frame, which we denote as a pulse packet. This is verified in the measured hydrophone data of Fig. 2c at different depths, in which the same Verasonics linear array probe is used with beam-forming configurations identical to the simulation conditions of Fig. 2b. The amplitude distribution of the pulses in a pulse packet is a strong function of the relative distance between the imaging probe and the implant of interest. This imperfect beam-forming is universal in all practical ultrasound imaging systems. As a result, motes within the FoV will find the ultrasound waveform different for the same downlink bit and backscatter a different number of pulses for the same uplink bit depending on its depth.

To ensure that the mote can establish a reliable data link when positioned anywhere within the FoV, the downlink data as transmitted by the imaging transducer is kept constant for the entire frame. At the mote side, backscattered data is also kept constant for each imaging frame. This eliminates any dependency data transmission may have on the exact shape of the pulse packet, since every pulse within each packet carries the same downlink or uplink bit. The data rate in this case is determined by the frame rate.

**System-level design.** Figure 3a shows the block diagram of the imaging system and the designed mote. The timing of the signals used to implement the physical layer is illustrated in Fig. 3b. Using standard imaging hardware, beam control is customized in the software to allow different pulse widths to carry the downlink data. On the mote, the rectifier must harvest energy from the imaging ultrasound, and the voltage regulator needs to use the rectified power to provide proper voltage regulation for the rest of the mote. Each mote in the field-of-view observes pulse packet at different times in the frame period depending on its lateral coordinate and receives different pulse packet shapes depending on its depth from the transducer array. In Fig. 3b, a pulse packet centered at the 74th scanline out of a total of 192 is illustrated as

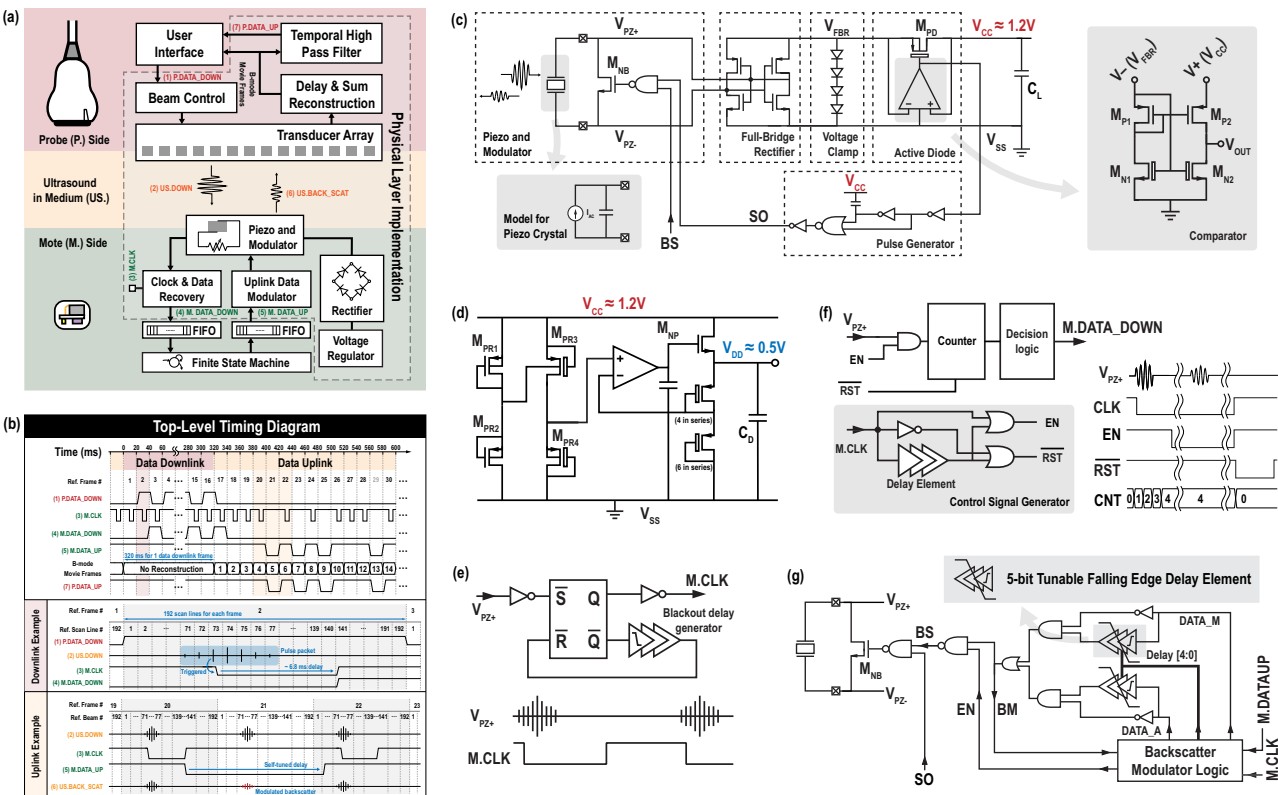

**Fig. 3 Circuit implementation of the physical layer. a** Block diagram for the imaging system and the mote with the physical layer implementation highlighted; **b** top-level timing diagram of the key signals, with the pulse packet seen from the mote, assumed between 71st and 77th scanline of each frame; **c** the active rectifier; **d** the voltage regulator; **e** the clock recovery circuit; **f** the data recovery circuit; and **g** the uplink backscatter modulator.

an example, corresponding to a mote found along the 74th scanline in the lateral coordinate. These received pulses are used to recover a clock synchronized to the frame rate, which is used for chip operation. This clock allows the mote to synchronously recover the downlink data and backscatter the uplink data; both are 1-bit digital signals for robustness. The backscattered ultrasound is picked up by the imaging transducer array as a part of the B-mode echo in the next frame, which, after standard delay-and-sum image reconstruction, forms the B-mode movie. The modulated uplink data thus translate to a frame-to-frame intensity change at the location of the mote. A software-defined temporal high-pass filter is used on the reconstructed B-mode movie to assist data visibility. A link-layer frame and an application-layer protocol, described in more detail below, complete the communication structure.

**Rectifier design and power regulation**. To harvest heavily duty-cycled ultrasound energy, we use an active rectifier (Fig. 3c) based on a previously published design[28] because of its superior on-off ratio. However, in this design, static power from biasing can significantly reduce the effective efficiency when operated at duty cycles at the level of ten ppm. In order to reduce the power consumption required to bias the continuous-time comparator operating at MHz-level switching speeds, we modified the original design with a dynamic biasing scheme, in which the gates of the current source pair $M_{N1}$ and $M_{N2}$ are connected to the inverting input of the comparator, which is connected, in turn, to the output of the full-bridge rectifier ($V_{FBR}$). In this way, the comparator is able to deliver a sub-μs decision time, with negligible static power consumption when no ultrasound energy is present.

Many previous efforts have relied on operating the piezoelectric transducer at the resonance of the piezoelectric crystal for

maximum power delivery[28,29]. However, this approach introduces two problems. First, a resonant power delivery requires a high-quality mechanical resonance, which, in turn, demands highly mismatched boundary conditions around the piezo transducer of interest. This can be achieved by adding an air-pocket-based backing structure underneath the piezo transducer[12], at the cost of a more complicated packaging as well as an increased mote volume. Second, significant impedance changes typically occur within the approximately 100-kHz bandwidth around the package-defined resonance frequency. As a result, the amount of power delivered strongly depends on the frequency tuning resolution of the imaging transducer array, making the motes difficult to power in practice.

For these reasons, here, we choose to assume a capacitive input impedance, a typical case when the piezo transducer is surrounded by non-optimized mechanical boundary conditions (see Supplementary Section 1). Resonance is still possible in the electrical domain with the help of an inductor for conjugate matching, but we choose not to do this because it would significantly increase implant volume. Instead, to boost the power harvesting without resonance, we employ a switch-only rectifier, a technique commonly used in low-frequency mechanical energy harvesting circuits[30] but adapted here for carrier frequencies in the MHz regime for reliable, nW-level power harvesting (see Supplementary Section 2). Such an approach consumes little on-chip area while relaxing the requirement for a high-quality mechanical resonance from the piezoelectric element. These benefits together enable efficient off-resonance power harvesting and the potential to scale to sub-0.1-mm³-range implant volumes with piezoelectric transducer integration[14,31].

The rectifier generates a 1.2-V dc supply, $V_{CC}$, which is clamped by the voltage limiter and is used to power circuits which

must have enough performance to track the MHz-level ultrasound carrier. A 0.5-V $V_{DD}$ is generated from $V_{CC}$ with a linear voltage regulator (Fig. 3d) for circuit blocks, such as delay generators and digital circuits, that require an accurate supply voltage but operate at lower performance requirements, typically synchronized to the 50-Hz frame rate, for dynamic power reduction. A pW-level trim-free voltage reference circuit[32] is used to produce a supply-independent and temperature-independent 0.5-V reference voltage from $V_{CC}$ for this linear regulator.

**Clock synchronization and bi-directional data telemetry.** Once sufficient energy is harvested for start-up, the mote establishes synchronized bi-directional data transmission with the imager. Figure 3e shows the circuits for clock recovery. When receiving a pulse that is higher than the threshold of the inverter, the set-reset (SR) latch produces the falling edge of the clock. An on-chip black-out delay generator with a nominal delay of 6.8 ms is inserted to delay the reset of the SR latch, effectively preventing the clock from being retriggered by pulses in the same pulse packet. The SR-latch reset triggers the rising edge of the clock. The inverter, as well as all other gates that directly connect to $V_{PZ+}$, are implemented with thick oxide devices and are further protected by on-chip electrostatic discharge (ESD) diodes to ensure voltage compliance.

To support the data downlink, it is necessary to measure the duration of one of the pulses in the pulse packet. Here we count the number of cycles in the first pulse that meet the threshold for clock recovery within each pulse packet (Fig. 3f). Decision logic then converts this count value into a 1 or 0 based on historical count values (see Supplementary Section 3).

The uplink data appear as a frame-to-frame change in intensity within the reconstructed image, which results from a change in the backscattered echo. To implement this back-scatter ASK, the circuits need to modulate the effective acoustic impedance of the transducer by changing the electrical load attached to it. Here, we choose to maximize the modulation depth by shorting the piezoelectric element through $M_{BS}$ (Fig. 3g) when transmitting an uplink bit 0. This approach creates three issues which must be addressed in the design. First, when the transducer is shorted, no clock generation occurs from the pulse packet in the given frame. This missing clock cycle provides no mechanism to synchronously recover from sending a 0. To rectify this, the backscatter modulator asynchronously resets any bit 0 to 1. This approach extends each bit 0 to take two clock cycles (see Supplementary Section 4 for details). To maintain a data-independent frame length, each bit 1 is also artificially extended to take two cycles, such that the uplink data rate is data-independent at one-half of the frame rate. Second, no downlink data can be recovered when an uplink 0 is transmitted; this precludes full-duplex operation. Third, the power harvesting is uplink-data dependent. As a result, the implant must be able to operate in the worst case at half the maximum power that would otherwise be harvestable for the case of a continuous string of 0's being sent in the uplink channel.

**Link layer and application layer.** To implement the digital control of the chip, a custom low-leakage standard-cell library was employed. To enable unique downlink communication to multiple implants within the same FoV, an identification generator is embedded in the application layer that assigns a unique identification sequence to each mote, implemented with an eight-bit fuse array. The frame structure of the link layer and the available instructions in the application layer are presented in Supplementary Section 5.

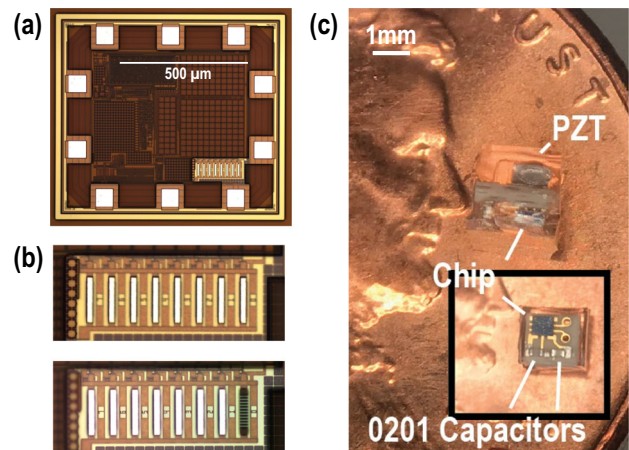

**Fig. 4 Fabricated IC. a** Die micrograph; **b** an example of a processed fuse; and **c** a fully integrated implantable mote.

**Post-processing and packaging.** The mote ASIC is fabricated using a TSMC 180 nm MSRFG 1P6M process, occupying a volume of $830 \times 740 \times 300\ \mu m$ (Fig. 4a). Fuses at the bottom right part of the ASIC are etched to assign an implant identification number different from the default value of 0xFF (Fig. 4b, an intact fuse indicates a bit 1). After this post-processing step, the chips are packaged into a fully integrated implant by adding two 100-pF decoupling capacitors that measure $0.6 \times 0.3 \times 0.3$ mm (0201, Murata Manufacturing Co., Ltd) for each of the $V_{CC}$ and $V_{DD}$ power domains ($C_{VCC}$ and $C_{VDD}$, respectively), as shown in Fig. 3a, and one $1 \times 1 \times 0.5$ mm lead zirconate titanate (PZT 5A, Piezo system) piezoelectric crystal (Fig. 4c) on a $2 \times 2 \times 0.66$ mm printed circuit board (PCB). Further size minimization is possible with monolithic integration[31].

**Performance characterization.** Initial electrical testing is performed to verify the functionality of the ASIC. To stay fully functioning, the chip requires 57 pW of static power when the dc output of the rectifier reaches 1.2 V. Achieving such ultra-low-power consumption allows for higher acoustic energy attenuation, supporting mote operation in deeper tissue.

For ultrasonography, a Verasonics Vantage 256 research ultrasound imaging systems is connected to an L12-3V (Verasonics Inc.), a 192-element linear array probe. 31 elements are selected and phased to generate focused beams at a 4.0323-MHz center frequency and 1-μs average pulse duration. In total, 192 beams are scanned on 0.2 mm steps in x direction at a 100-μs interval, generating a 38.6 mm image width. For each beam, backscatter is recorded for 1280 samples at 15.625 MSamples/s, resulting in an 81.9-μs recording time. For a sound speed of $c_s = 1540$ m/s, this gives a 63.1 mm maximum image depth. An 800 μs wait time is inserted between frames to create an imaging session running at 50 fps. For initial characterization, the motes are immersed in castor oil (Sigma Aldrich) that has an attenuation factor of 7.21 dB/cm at 4 MHz[33], to better emulate acoustic power attenuation in lossy tissue. A custom-implemented delay-and-sum algorithm is used in image reconstruction for uplink data identification (see Supplementary Section 6). Figure 5a shows the experiment setup.

To harvest sufficient power and deliver detectable uplink modulation depth, the mote requires the focused pulse amplitude in a pulse packet to reach approximately 400 kPa, at which point each pulse provides an input energy of approximately 52.8 nJ. Motes are able to start up after power regulation is complete, which typically requires about 235 frames or 4.7 s (see Supplementary Section 7). In the reconstructed floating-point-

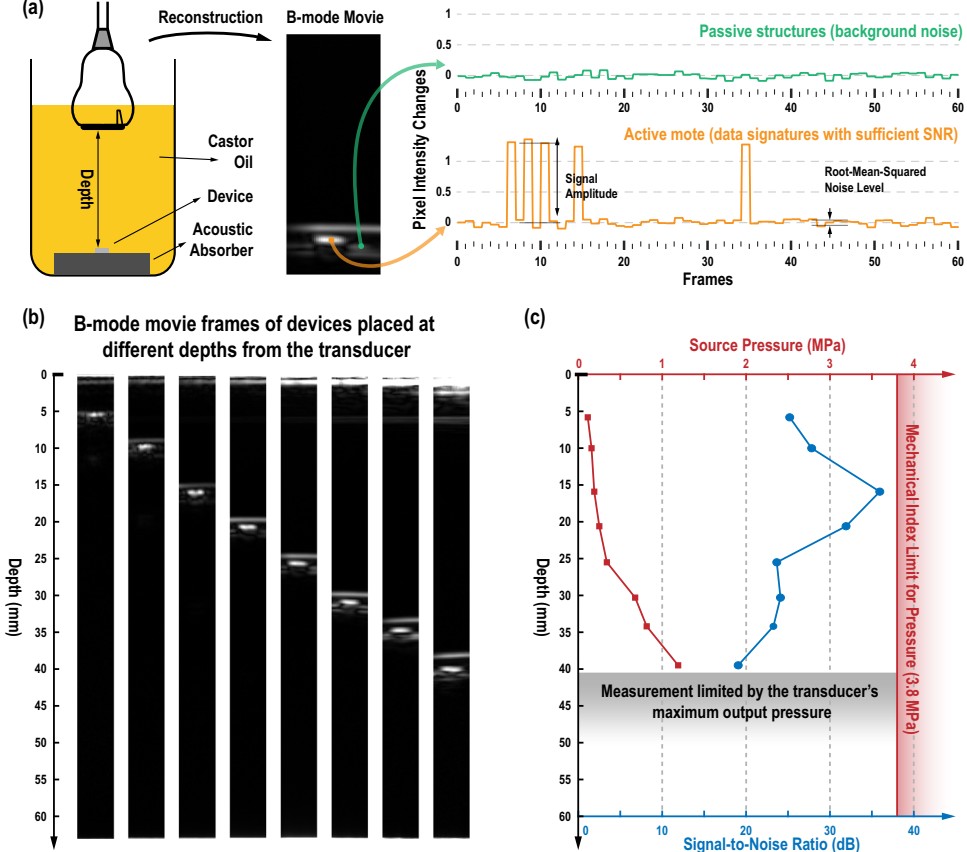

**Fig. 5 Characterization of the mote in a controlled ultrasound environment. a** Experimental setup and pixel intensity changes at passive structures and the active mote across frames in the reconstructed B-mode movie; **b** B-mode movie frames for depth characterization; and **c** the minimum source pressure required to properly power the mote at these depths and the detected signal-to-noise ratio in the corresponding B-mode movies.

valued B-mode images, the uplink data sent from the mote leads to a significant, regular brightness change, which aids in mote localization. The ratio between the signal power and the noise power, or signal-to-noise ratio (SNR), varies as the mote moves away from the source transducer, shown qualitatively in Fig. 5b, c. With a fixed 15-mm focal depth, the mote stays fully functioning with an uplink SNR of at least 19.1 dB up to 40 mm away from the transducer array. The SNR for the uplink data as rendered in the reconstructed image is higher than that produced by measuring the echo at a single transducer because the delay-and-sum algorithm utilizes the spatial correlation of echo data receive from multiple elements. The 40-mm depth is practically limited by the maximum voltage that can be applied to the piezoelectric elements on the L12-3V probe to generate ultrasound pressure waves. Still, this maximum pressure is much lower than the safety limit for human use as regulated by the mechanical index (MI) threshold of 1.9 kPa/$\sqrt{\text{Hz}}$[10], or 3.8 MPa at 4 MHz, beyond which tissue damage can occur. The corresponding ultrasound power is only 9.2 mW/cm$^2$, much lower than the 720 mW/cm$^2$ safety limit[10,12].

**Ultrasonography with phantom**. To demonstrate the functioning of the motes within an imaging application, chicken breasts are hand-sliced into about 1-cm-thick layers and soaked in castor oil to act as a tissue phantom, with two motes between the chicken-breast layers (Fig. 6a). For interference-free operations, the motes are separated by at least the beam width of 1 mm and are not stacked vertically to avoid the top mote blocking the backscattered data from the bottom one.

Figure 6b shows the B-mode image captured by the linear array transducer. High-pass temporal filtering of the B-mode movie (see "Methods") is used to amplify the presence of the motes. Once the motes' locations are known, localized intensity changes are analyzed to extract the uplink data. In this case, the transducer array first sends out a QUERY_ID instruction (see Supplementary Section 5), to which all the motes respond by sending their specific identifiers (IDs) back (Fig. 6c). This instruction is designed to be non-device-specific, a fast way to track all the active motes implanted within the FoV, which also enables direct instruction addressing to a specific mote. To verify device-specific operations, the simplest identification-sensitive instruction HELLO is sent out to each of the devices, with the just-captured ID as the argument of the instruction (see Supplementary Section 5). Interference-free replies are then captured from the received echoes. It is worth noting that in this case, it is difficult to visually identify the location of the motes directly from the B-mode movie, as this miniaturized implant is not easily distinguished from other structures in this complicated mechanical environment. However, as active devices, they present temporal modulation far stronger than the baseline noise level, with a greater than 23.4-dB SNR from the mote placed 15.1 mm away from the transducer, and a greater than 22.1-dB SNR from the mote placed 26.5 mm deep, for all received uplink data. This property allows easy location of multiple motes, as well as robust communication with multiple motes simultaneously, as shown in Fig. 6c and Supplementary Movie 1.

**In vivo demonstration**. To demonstrate the functioning of the mote in vivo, the devices are implanted into the lower hindlimb of a mouse, while the same B-mode imaging system and imaging

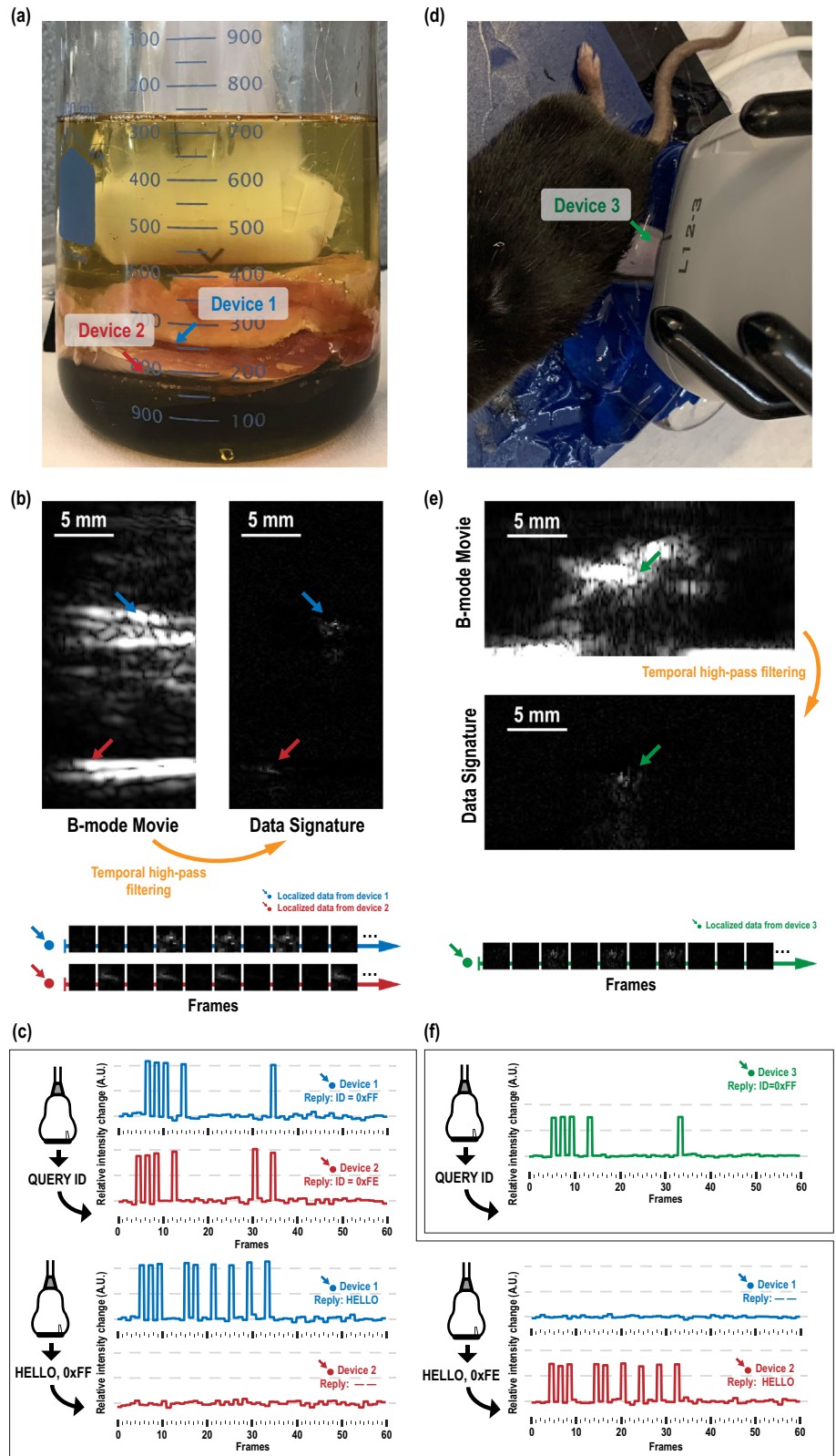

**Fig. 6 Measured response from the mote in both in vitro and in vivo environments. a** Setup with two devices embedded in layers of chicken breast, with approximate mote locations overlaid; **b** the B-mode movie captured by the imaging system, where temporal average removal is used to reveal locations with abrupt, frame-to-frame intensity changes (plotted versus frames at the bottom); **c** detailed timing diagram showing instructions sent from the transducer and the motes' replies (in frame-domain, a detected change implies a bit 0, information translated according to Supplementary Table S3/S4); **d** a setup with one devices implanted in the mouse's hindlimb; **e** the B-mode movie and data pattern from the mote for this in vivo experiment; and **f** detailed timing diagram showing the interaction between the imaging system and the implanted mote.

configuration used for the phantom experiment is used to communicate with the mote (Fig. 6d). An acoustic absorber is placed below the hindlimb to reduce excess ultrasound reflection from the thermal pad underneath, mimicking imaging in a larger animal in which ultrasound power attenuates beyond the field-of-view. The B-mode movie of the hindlimb at 50 fps is shown in Supplementary Movie 2, with tibia and fibula clearly identified and the mote located in the top middle of the image. Modulated backscatter is evident at the location of the mote. By overlaying the extracted data signature within the original B-mode movie, the motes can be tracked in a biogeographic aware fashion; that is, not only can we retrieve the absolute location of such devices, but we can also determine its location relative to surrounding structures. The SNR for this data exceeds 28.5 dB from the mote placed 10.8 mm away from the source linear transducer.

**Conclusion**. Here, we have presented the design of the implantable mote that operates in a manner directly compatible with widely adopted medical ultrasonography, opening up new possibilities for next-generation biosensing implants that are miniaturized, battery-less, distributed, real-time, and trackable in a biogeographically aware fashion. Such motes could also find applications in real-time surgery site tracking[26]. All of these applications constitute a new approach for augmented ultrasonography.

## Methods

**Mote design and packaging**. The integrated circuit is fabricated in a 180-nm 1P6M salicide CMOS process with AlCu FSG back-end metallization (Taiwan Semiconductor Manufacturing Company). The chip measures 830 µm by 740 µm in area and is 300-µm thick. A 2-mm-by-2-mm-by-0.66-mm PCB is designed with electroless nickel electroless palladium immersion gold (ENEPIG) finish to integrate all the required components for the implantable mote. Gold stub bumping is used to attach the chips to the PCB. To assemble the mote, all the pads on the chip are first thermosonically attached with gold studs using a wirebonder and then bonded to the PCB using the Fineplacer Lambda tools (Finetech GmbH & Co. KG) with thermocompression face-bonding[34]. The package side is heated to 340 °C for 10 s; while the chip is heated to 260 °C for 20 s. A compression force of 4 N is used. Two 0201 capacitors at 100 pF each (Murata Manufacturing Co., Ltd) are soldered using silver epoxy H20E-D (Epoxy Technology Inc.), and the 1-mm-by-1-mm-by-0.5-mm PZT 5A (Piezo Systems) transducer is mounted on the PCB. Finally, the fully assembled mote is encapsulated using polydimethylsiloxane (PDMS, Silgard 184 Silicon Elastomer, Dow) as a passivation layer.

**Fuse etching**. To selectively etch fuses corresponding to specific bits of the ID generator, chromium photo masks (Telic Company) are processed using mask writer (Heidelburg Instruments) with an opening that aligns to one exposed fuse on the chip. Each chip is mounted on 500-µm-thick silicon (University Wafers) carrier, and spin-coated with AZ1813 positive photoresist (Microchemicals). Arbitrary ID assignment is performed by repetitive exposure using an MA6 mask aligner (SUSS MicroTec) on each fuse that needs to be etched. Patterned chips are then submerged in ferric chloride (FeCl₃, Transene Electronic Chemicals) etchant for fuse removal. The chips are then soaked in a mixture of 10% ammonium hydroxide (NH₄OH, Sigma Aldrich) and 20% hydrogen peroxide (H₂O₂, Fisher Scientific) for one minute to remove the seed layer between the dielectric passivation and the fuse.

**Modified ultrasound imaging**. To perform the ultrasound-related experiments, the Verasonics Vantage 256 research ultrasound system with a L12-3V linear array probe (Verasonics Inc.) is customized through its built-in MATLAB (Mathworks Inc.) interface. For the ultrasound imaging session, each frame consists of 192 scan lines, or ray lines, spatially separated by 0.2 mm, with a 100-µs interval between adjacent scan lines. This gives the B-mode image a width of 38.6 mm. To focus the beam, 31 elements in the linear array are selected (less if the scanline is too close to the edge), and the focal depth is used to calculate the pulsed delay at each element. An 800-µs delay is inserted between frames to maintain a frame rate of 50 fps.

The pulse width is kept constant for all of 192 scan lines in a frame. In frames with downlink data transmission, either a three-cycle-long (0.75 µs) pulse, or a five-cycle-long (1.25 µs) pulse is used to represent downlink bits 0 and 1, respectively. When waiting for uplink frames from the implants, four-cycle-long (1 µs) pulses are used for stable power delivery on the transmit side, while up to 127 elements are used to collect the reflected echo pressure for B-mode image reconstruction. Reflected ultrasound data are collected at 15.625 MSamples/s through Verasonics' built-in hardware for 1280 samples per scanline per receive element, resulting in an 81.9 µs recording time for up to 63.1 mm imaging depth, assuming the media has a sound speed of $c_s = 1540$ m/s.

Before the experiment, the ultrasound energy from the linear array transducer is captured and calibrated using a hydrophone with ultrasound clear transmission gel (Aquasonic) as an acoustic media, allowing the measurement of the pulse packet's energy distribution as well as the calibration of the absolute pressure. Such waveforms are available as Supplementary Fig. S4. From the measured beam profile, the minimum possible lateral resolution is about 1 mm at the focal depth. A 1-µs pulse duration gives an axial resolution around 1.5 mm.

**Image reconstruction**. A delay-and-sum algorithm[27,35,36] is first used to reconstruct the B-mode image with floating-point intensity values. A digital Gaussian filter, with a center frequency equal to the transmit frequency (4.0323 MHz) and a bandwidth of 100 Hz, is then applied to the reconstructed image along the depth direction, followed by an envelope detection algorithm to improve the quality of each image. To detect the uplink data, we perform a temporal high-pass filtering of the image by performing window-and-average over 60 consecutive B-mode frames and subtracting this average image from each frame. This effectively removes motion artifacts lower than 0.8 Hz, yet leaves the data modulation occurring at the frame rate of 50 Hz unaffected. To further improve the visibility of the uplink data, the data are re-scaled such that the maximum intensity becomes 255, the maximum intensity supported for eight-bit displays. Random scattering of ultrasound recorded from the probe, after reconstruction, is treated as the frame-to-frame noise floor when characterizing the reliability of the data in the form of SNR. Reconstructed images are discarded for frames containing downlink data, as a frame-to-frame change in pulse width produces imaging artifacts.

**Tissue phantom experiment**. The tissue phantom was prepared using boneless chicken breasts that were hand-sliced into roughly about 10 mm thick layers. An acoustic absorber (F28, Precision Acoustics) was placed at the bottom of a 1 L beaker as the backing layer, and two layers of chicken breasts are placed on top of it. Castor oil (Sigma Aldrich) is then used to fill the beaker as the default acoustic medium. One of the devices is placed on top of the backing layer, while the other is placed between the two layers of chicken breasts. The L12-3V linear array probe is then placed on top of the chicken-breast layers, without touching the tissue. A center frequency of 4.0323 MHz is chosen from the Verasonics Vantage System's supported frequency list, as a trade-off point between transmission attenuation and image quality. The linear array probe is fixed by a chemical stand, and powered at 26 V driving voltage (calibrated to ~650 kPa) with a focus of 25 mm, running the modified ultrasound imaging program, polling the two devices within the same field-of-view.

**Surgical procedure**. The Institutional Animal Care and Use Committee (IACUC) reviewed and approved protocols for Columbia University's program for the humane care and use of animals and inspects the animal facilities and investigator laboratories. Evaluation of the implanted devices was performed in compliance with Animal Welfare and Columbia's IACUC regulations under the approved IACUC protocol. C57BL/6J (stock number 000664) mice were obtained from Jackson Laboratories and housed in Columbia's ICM facility. Surgeries were performed in animals 12–24 weeks old.

Mice were anesthetized with 1 g/kg urethane administered intraperitoneal and placed on a heating pad lying on the ventral side. The left hindlimb and palm were fixed facing up to a base by means of cyanoacrylate glue (Krazy Glue) to minimize movement caused by breathing. With the posterior region of the limb facing up, the skin and underlying tissue were cut open with fine scissors to reveal the gastrocnemius muscle at an approximate distance of 1 cm from the Achilles tendon. The device was placed between the skin and the muscle, without further fixation. The skin was closed, covering the device.

**In vivo experiments**. The mouse was placed on an acoustic absorber (Aptflex F28, Precision Acoustics) with the hindlimb fixed by means of a cyanoacrylate glue after surgery. A heating pad was placed under the absorber to assist in maintaining the animal's body temperature. The L12-3V linear array ultrasound probe (Verasonics Inc.) was fixated by a chemical stand, and placed on top of the hindlimb without touching the hindlimb itself. An ultrasound-clear transmission gel (Aquasonic) was applied over the hindlimb to fill the gap between the probe and the muscle, serving as the default acoustic medium. The ultrasound program identical to that described in section Tissue Phantom Experiment is used to communicate to the implanted device.

## Data availability

The ultrasound images generated in this study are available at https://github.com/klshepard/imaging_motes. All other relevant data are available from the corresponding author upon reasonable request.

## Code availability

The MATLAB code for image reconstruction and mote detection based on the Verasonics Vantage System's output is available at https://github.com/klshepard/imaging_motes.

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

## Acknowledgements

This work was supported in part by the Defense Advanced Research Projects Agency (DARPA) under Contract HR0011-15-2-0054 and Cooperative Agreement D20AC00004.

## Author contributions

Y.Z. and K.L.S. conceptualized the study. Y.Z. designed the circuits. Y.Z., P.M., V.A.-P., I.U., and J.E. performed the experiment. Y.Z. and K.L.S. wrote the manuscript. K.L.S. provided overall supervision and guidance. All authors provided active and valuable feedback on the manuscript.

## Competing interests

The authors declare no competing interests.
