## [Peer Review File · Nature Communications]

Reviewers' Comments:

Reviewer #1:

Remarks to the Author:

The concept of augmented ultrasound is a great application for ultrasound motes. The manuscript is well-written and easy to follow.

General comments:

1. What's the limiting factor on volume (10mm³)? Most ultrasound motes, including examples from these authors, are much smaller in volume. In particular, the manuscript claims that the proposed approach allows the piezo to scale to much smaller volumes.
2. How do the motes interfere with the intended image? Since they reflect ultrasound (even when not modulated), they likely interfere with the surrounding image. Are the imaged reflections larger than the volume of the mote? This would potentially cause interference with the image over a larger volume. The movies provided are not at high enough resolution to allow the reader to infer this information. There should be some quantitative assessment of the volume of surrounding tissue and tissue below the mote at higher depths where imaging SNR would be degraded.
3. How do the wand modifications impact the images?
4. Why were the sensors not included? Given the prior art in this group it would have been fairly straightforward to implement, was there an issue or limiting factor?

Specific comments:

5. "In this focused-beam approach, accurate knowledge of the implant location is required for power delivery 50 and communication." Some prior art uses focused beam, while others use unfocused transducers to mitigate the location issue. Please double check the prior art cited (5, 10, 12-13) for focused vs. unfocused transducers since I believe not all four of the papers you cite use solely focused beam.
6. "Some efforts have been directed toward achieving real-time localization using high order harmonics from the implant's reflected acoustic waves" There are other papers that you do not cite that address this issue, such as:
Meng, et al. "Self-Image-Guided Ultrasonic Wireless Power Transmission to Millimeter-Sized Biomedical Implants"
Benedict, et al. "Time reversal beamforming for powering ultrasonic implants"
I don't think yours is the only solution to this problem, but your system is advantageous since it integrates with imaging. Please revise this section.
Some of this prior art uses 2D arrays of transducers, which would give the ability to focus in 3D. Your device is a linear array, and therefore you can only focus in 2D. How do you deal with movement in the 3rd dimension? Would the user have to find it by moving the wand?
7. Line 60 "allowing the probes to be biogeographically located in the image" do you mean motes?
8. "in which 31 elements at a time are phased to produce a focused, z-directed pulse at a given position in x along the transducer array, which is then scanned to produce a frame." Can you give more detail about how this calibration and computation is done? How do you deal with scattering?
9. "This imperfect beam forming is universal in all practical ultrasound imaging systems." While this is true, in an imaging system the wand is completely freely moving relative to the body since it is handheld. Additionally, since the output power is quite limited, alignment may be more of an issue. How would this be managed in a practical imaging system where the wand is handheld? Do you have any trials where the wand is handheld or are all tests done with a fixed wand location?
10. Operating off-resonance and using a switch only rectifier is an interesting idea, and the supplementary information is helpful. I don't think resonant power delivery requires a high Q (your spec is ~40), but I agree that air-backing reduces the internal resistance of the piezo and ultimately improves the harvested power. Most ultrasonic motes have moderate quality factors which would not cause such large shifts with small perturbations in resonant frequencies.
11. There are many highly efficient switching techniques used in low-frequency piezo power harvesters where the piezo is best modeled by a capacitor and a current source. How does the efficiency of this chosen technique scale to the MHz range? How does it compare with a resonant device in terms of absolute power harvested? I can see that it is qualitatively more efficient from your diagram, but a comparison of volume vs. harvested power would be very interesting.
12. The design of the active diode doesn't need to be so prominent in the text since this topology is well-known and was used in ultrasound in Charthad '15.

13. "and has the potential to scale to sub-0.1-mm³ range implant volumes with piezoelectric integration" Why did you not scale to a smaller volume if this is possible?
14. Font sizes in Figure 2 should be increased.
15. Figure 3 is low resolution and hard to read.
16. In Fig. 3 you show Pz+ going directly into a logic gate. Is there no issue with this?
17. Fig. 3 would benefit from a higher level signal timing diagram.
18. In Figure 4, what are the dimensions of the PZT and of the chip? Can you please add measurement bars?
19. In Figure 5 mechanical index is given at each depth, which is great. Can you also provide intensity values since that is also an important limit for ultrasound?
20. In Figure 6 it would be helpful to align the ultrasound image with the high pass image so that it is visually easier to see the location of the motes in the former. Alternatively, the same locations can be labeled in the former.
21. In Figure 6b there are bright spots surrounding the mote in the high pass image. They are under the circle and therefore hard to see. What are these caused by? Can you move the circle so that these can be seen?
22. You are using an acoustic absorber for in-vivo measurements. If this is required, how do you anticipate the device can be used in a more realistic environment?
23. The movies are a nice and helpful demonstration. For device 3 in the second movie, the final "0" lights up in a different location. What causes this?

Reviewer #2:

Remarks to the Author:

This is solid, novel work but in our opinion, requires additional work to properly substantiate the claims made.

-A single implant was imaged at a maximum depth of 40 mm in a homogenous medium. In the abstract, the authors claim that an arbitrary number of motes can be imaged within a field of view and that signal-to-noise ratios of 19.1 dB are demonstrated at depths of up to 40 mm in tissue. Additional data is needed to verify this claim in more realistic tissues where multiple motes are implanted.

- We believe that some of this work was previously published in the 2019 IEEE Custom Integrated Circuits Conference. The authors should clarify what is new vs. what has been published.

Introduction

-The authors may consider more broadly discussing the state-of-the-art wireless systems based on miniaturized implants. The key advance of this work, namely the wireless power/communication capabilities that enable postsurgical localization of the implants, should be set in the context of the prior body of work in bioelectronics. See, for instance:

S. Sonmezoglu, et al. "Monitoring Deep-Tissue Oxygenation with a Millimeter-Scale Ultrasonic Implant," *Nature Biotechnology* 39, 855–864 (2021).

Kim, Albert, et al. "An Implantable Ultrasonically-powered Micro-Light-source (μ Light) for photodynamic therapy." *Scientific Reports*, 9.1 (2019).

- The authors do not review existing technologies used for determining the location of miniaturized implantable medical devices inside the body. The authors should compare their localization approach based on ultrasound imaging with the current techniques based on, for example, radio frequency (RF) signals, magnetic fields, or nuclear magnetic resonance.

Design considerations

- The power transfer & data communication protocol and the reconstruction algorithms used to form an image are not clear from the treatment in the main manuscript. We recommend adding a timing diagram to demonstrate the power and downlink/uplink data waveforms and simplified block diagrams to explain decoding of the backscattered signals to form an image.

- Is the frame rate dependent on the operating depth of the mote due to time-of-flight? For this imaging system, is the operating depth of the mote limited by the fixed frame rate of 50 fps? Please clarify in the text.

- It is not clear why the authors chose to use two cascaded rectifiers, for example, instead of a single active full-wave rectifier? The authors are expected to characterize the rectifiers and report the related data. Is the voltage clamp used to limit the harvested voltage to 1.2V?

- Can the authors provide the measured impedance of the packaged piezoelectric crystal versus frequency in castor oil and indicate the operating frequency and impedance on the figure? Is there a significant impedance mismatch between the piezo crystal and input to the full-wave rectifier at the expected operating voltage? How much can the power harvesting efficiency be improved through impedance matching?

- The authors state that "First, a resonant power delivery requires a high-quality mechanical resonance, which, in turn, demands highly mismatched boundary conditions around the piezo transducer of interest... Second, significant impedance changes typically occur within the approximately 100-kHz bandwidth around the package-defined resonance frequency."

- We disagree with this statement. The size of a high-quality factor piezoelectric crystal can be adjusted to achieve impedance matching between the piezo and the input of the rectifier, depending on the application. Furthermore, a high-quality factor piezo achieves significantly higher modulation depth at two extreme piezo terminations (short- and open-circuited piezos) at resonance, improving the uplink data reliability of the system. Can the authors explain why significant impedance changes typically occur in practice?

- Can the authors clarify how to boost the power harvesting without resonance by employing a "switch-only" rectifier in the chip? Furthermore, the authors mention that "Such an approach requires little complexity in packaging..." Does complexity in packaging mean that this approach requires the piezoelectric crystal to be closely integrated with the chip to reduce the electrical parasitics and hence to improve the power harvesting performance?

- The authors state "We choose to maximize the modulation depth by shorting the piezoelectric element through MBS (Fig. 3f) when transmitting an uplink bit 0." The switch (transistor) has an internal resistance when it is turned on to transmit an uplink bit 0. Can the authors provide the switch-on resistance, the electrical load resistance when the switch is off, and the measured backscatter amplitude modulation at the system operating frequency?

- The mote powers up when it receives the first ultrasound pulse packet and modulates the backscatter in the second ultrasound pulse packet, which limits the data rate to one-half of the frame rate. Each pulse packet is 1- μ s long. The 1- μ s packet duration seems very short to transmit uplink data from the implant when a sensor is integrated into the mote to monitor biomarkers, such as oxygen and pressure. With respect to the data transfer method, does this system only support analog backscatter amplitude modulation to transmit uplink data as the uplink data appears as a frame-to-frame change in intensity? Would it be possible to use digital backscatter amplitude modulation in this imaging system if multi-bit digital data from the implant sensor is desired to be transmitted to the external transceiver?

Readout from multiple motes

- It seems that this imaging system interrogates multiple motes in a time-division multiplexing fashion. In this approach, when many motes are implanted into different locations in a tissue of interest, the frame rate will be limited by the number of motes. What is the maximum number of motes that can be implanted into tissue for the frame rate of 50 fps? Would it be possible to enable simultaneous operation of the motes implanted in different locations of target tissue? Ultrasound reflections from the multiple motes in heterogeneous tissue at similar depths may interfere with each other at the face of the transducer array in a time interval within when the uplink data is received; this may degrade system performance. The operation of the system with multiple motes, at least two motes, at similar depths in explanted animal tissue should be reported.

- One of our concerns is the long-term reliability of the uplink data transmission between the implant and the external probe, introducing errors in the localization of the implants. How would the system localization accuracy be affected by changes in the angular position of the implanted mote due to external pressure, breathing, or scar formation?
- If multiple motes are closely implanted into different locations in a tissue of interest, ultrasound reflections from the motes may interfere with each other: this may degrade the system localization performance. How close can the motes be positioned in the transverse (x- and y-) and longitudinal (z-) directions in tissue?
- How would the localization accuracy of the system change with the field-of-view size in heterogeneous tissue?
- An ultrasound path with higher acoustic attenuation due to absorption and scattering can significantly degrade system data transfer reliability and hence system performance. The localization performance of the system with multiple motes at different locations in explanted tissue with skin and fat content, such as porcine muscle tissue, should be quantified and reported to demonstrate the effect of acoustic attenuation on the system performance.
- The modulation depth in an in vivo experiment is generally expected to be lower than the modulation depth measured in a homogeneous medium (castor oil), since the ultrasound reflections from internal tissue interfaces likely interfere with the ultrasound reflections from the motes. Does the modulation depth of the backscattered signal depend on the background scattering by the tissues? How does it compare to the castor oil experiment in Fig. 5? What is the maximum modulation depth achieved in castor oil, and what is the minimum detectable modulation depth by this system?
- It seems from Figs. 5 and 6 that the image quality obtained in the in vivo experiment is seen to be lower compared to that obtained in the in vitro and ex vivo experiments. Can the authors provide the in vivo localization accuracy and explain the reasons for degradation in the image quality in the in vivo operation?
- Can the authors provide estimated depths for the motes operated ex vivo and in vivo in Fig. 6? This can be done, for example, using the time-of-flight information (that is, the time delay between the received backscatter signal and the signal that drove the probe).
- The in vivo configuration is based on ultrasound transmission through layered tissues (skin, muscle, bone). The effect of reflections from this layered structure on the system performance should be quantified, as reflections from interfaces at different depths will result in multipath interference that may degrade performance. A possible way is to compare the localization accuracies obtained in the in vitro and in vivo experiments with the motes operated at similar depths.
- The in vivo experiment was performed with a single mote implanted into the lower hind limb of an anesthetized mouse. The multi-mote in vivo operation should be demonstrated to prove the system's capability of localizing more than one mote in layered tissues.
- What is the frequency of motion artifacts caused by, for example, breathing that the system can tolerate?

RESPONSE TO REVIEWERS

Reviewer #1 (Remarks to the Author):

The concept of augmented ultrasound is a great application for ultrasound motes. The manuscript is well-written and easy to follow.

General comments:

1. What's the limiting factor on volume (10mm³)? Most ultrasound motes, including examples from these authors, are much smaller in volume. In particular, the manuscript claims that the proposed approach allows the piezo to scale to much smaller volumes.

Response: In our previous work, we fully integrated the piezoelectric transducer onto the integrated circuit, which produces the smallest form factor, but requires extensive post-processing of the CMOS die to do this integration. Here we instead use miniaturized printed circuit boards, which makes the packaging simpler since little post-processing is now required, although the smallest form factors are not achievable. The minimum possible volume for mote packaged this way is limited by the volume of the constituents, excluding the carrier board. In our current approach, the mote needs 1.0×1.0×0.5 mm³ for the piezo transducer, 0.76×0.85×0.3 mm³ for the IC, and 0.60×0.30×0.30 mm³ for the 0201-packaged capacitors. This yields a minimum volume of 0.80 mm³. Currently, the carrier board measures 2.0×2.0×0.66 mm³. We expect that are this packaging approach could be reduced to the sub-10-mm³ range, but more aggressive form factors will require monolithic integration. We note this now in the manuscript in “Results – Post Processing and Packaging” and “Methods – Mote Design and Packaging”.

2. How do the motes interfere with the intended image? Since they reflect ultrasound (even when not modulated), they likely interfere with the surrounding image. Are the imaged reflections larger than the volume of the mote? This would potentially cause interference with the image over a larger volume. The movies provided are not at high enough resolution to allow the reader to infer this information. There should be some quantitative assessment of the volume of surrounding tissue and tissue below the mote at higher depths where imaging SNR would be degraded.

Response: The mote appears as a highly reflective rigid cube in the ultrasound image, since the PZT-5A crystal used in the mote shows a higher acoustic impedance than that of the bone (~36 MRayls for PZT-5A compared with ~8 MRayls for bone). We have added Supplementary Section 8, which shows k-Wave simulation of the resulting

shadowing of the mote using beam forming parameters identical to what we have in the experiments of Fig. 2. In this case, we model the mote as a 1.0-mm-by-1.0-mm-by-0.5-mm cube of PZT-5A. The resulting ultrasound B-mode images are summarized in Fig. S9, with motes placed at different distances apart from the source transducer, showing the interference to the image from the mote placed in the field-of-view. In the lateral dimension, the image of the mote is a spatial convolution between the lateral beam pattern and mote itself, reflecting the available lateral resolution. In the vertical direction, reverberation from the mote generates multiple copies of the motes image, that is strong up to 3 mm below where the mote is placed. The mote also shadows part of the ultrasound energy in the region below. However, since the focused ultrasound beam has a finite width in both the lateral and elevational direction, the shadowing is only significant up to 5 mm below the mote. It is worth noting that, in Supplementary Section 8, all simulations are done using specifications in our designed mote. The reported values here vary with changes in the beam forming parameter or the form factor of the mote.

3. How do the wand modifications impact the images?

Response: The modifications to the way imaging are done to accommodate the motes stems from the use of pulse-width modulation for the data downlink. In our current practice, all frames that are modulated for the data downlink are removed from the movies to avoid imaging artifacts, resulting in the loss of 16 frames (a 0.32-second pause in a 50-fps imaging session) whenever a data downlink frame is sent. This is now explained more carefully in the “Image Reconstruction” section within the “Methods”.

4. Why were the sensors not included? Given the prior art in this group it would have been fairly straightforward to implement, was there an issue or limiting factor?

Response: No, there are no issues. We are doing precisely this in ongoing future work.

Specific comments:

5. “In this focused-beam approach, accurate knowledge of the implant location is required for power delivery and communication.” Some prior art uses focused beam, while others use unfocused transducers to mitigate the location issue. Please double check the prior art cited (5, 10, 12-13) for focused vs. unfocused transducers since I believe not all four of the papers you cite use solely focused beam.

Response: We appreciate the reviewer for pointing this out. Some of the work cited here uses semi-continuous wave operation of single transducer. We have modified the corresponding text so that it better reflects the cited work.

6. “Some efforts have been directed toward achieving real-time localization using high order harmonics from the implant’s reflected acoustic waves” There are other papers that you do not cite that address this issue, such as: Meng, et al. “Self-Image-Guided Ultrasonic Wireless Power Transmission to Millimeter-Sized Biomedical Implants” Benedict, et al. “Time reversal beamforming for powering ultrasonic implants”. I don’t think yours is the only solution to this problem, but your system is advantageous since it integrates with imaging. Please revise this section. Some of this prior art uses 2D arrays of transducers, which would give the ability to focus in 3D. Your device is a linear array, and therefore you can only focus in 2D. How do you deal with movement in the 3rd dimension? Would the user have to find it by moving the wand?

Response: Thank you very much for the suggestion. We have revised the corresponding section in our manuscript to include these additional references for their ability to adjust delays in an array for proper focusing. For focusing, however, it remains true that one must have approximate knowledge of mote location; that is, it is necessary to move the probe around to find the mote. Our compatibility with the imaging process, however, significantly relaxes the required accuracy in locating the mote from several times the beam width to the full field of view of the imager. 1D arrays are usually implemented using elements that are longer in the elevational direction, giving a wider dimension to the beam in this direction, which can aid in locating the mote. This finite beam width in the elevational direction provides some tolerance to movement along the third dimension. Within our experiment, we need to scan the wand initially to find the mote and orient the wand such that movement is largely constrained in the imaging plane. This process, if implemented with 2D-array transducers, will require little to no mechanical scanning because the 2D array can access a full 3D volume, as noted by the reviewer. This has now been clarified in the Introduction in the manuscript.

7. Line 60 “allowing the probes to be biogeographically located in the image” do you mean motes?

Response: The authors thank the reviewer for pointing out. Yes, we meant motes. This has been corrected.

8. “in which 31 elements at a time are phased to produce a focused, z-directed pulse at a given position in x along the transducer array, which is then scanned to produce a frame.” Can you give more detail about how this calibration and computation is done? How do you deal with scattering?

Response: In the “B-mode Sonography and Challenges in the Physical Layer” section, we aim to provide a qualitative description of how medical B-mode ultrasound operates in the context of the challenges in designing the mote. For clarity, we use the imaging

process that is used later for *in vitro* and *in vivo* testing of the mote as an example. The calibration and computation process are exactly the same as those described later in the “Results” and “Methods” sections. This is now clarified more completely in the text. Scattering is part of the B-mode image reconstruction. We remove two scattering sources which are not part of the image. The first is the initial scattering at the boundary between the probe and the media, creating a bright region around top part of the resulting image. This is removed in the final image. The second is the scattering at the bottom of the media of interest, which is reduced by a piece of acoustic absorber placed at the bottom of the media.

9. “This imperfect beam forming is universal in all practical ultrasound imaging systems.” While this is true, in an imaging system the wand is completely freely moving relative to the body since it is handheld. Additionally, since the output power is quite limited, alignment may be more of an issue. How would this be managed in a practical imaging system where the wand is handheld? Do you have any trials where the wand is handheld or are all tests done with a fixed wand location?

Response: In the cases reported in the manuscript, the imaging transducer is at a fixed location such that the data link is stable. However, we do move the transducer around before fixing it when initially searching for the mote. A stable datalink from a fixed wand location gives us the ability to process the data from the mote without having to do this in real-time. Further enhancement to the data collection infrastructure will enable real time operation, which would allow communication with the motes without fixing the imaging transducer. We have clarified this in both the “Tissue Phantom Experiment” and the “In vivo Experiments” sections in the “Methods”.

10. Operating off-resonance and using a switch only rectifier is an interesting idea, and the supplementary information is helpful. I don’t think resonant power delivery requires a high Q (your spec is ~40), but I agree that air-backing reduces the internal resistance of the piezo and ultimately improves the harvested power. Most ultrasonic motes have moderate quality factors which would not cause such large shifts with small perturbations in resonant frequencies.

Response: Here, our concern with using a resonance-based rectifier derives mainly from having to contend with the varying impedance of the piezoelectric element. Although the reviewer is correct that the mote can operate with a moderate quality factor, the exact quality factor of the piezo becomes a function of its mechanical load. The IC in the mote is designed to be compatible with monolithic integration, and we would like to relax requirements on the mechanical boundary condition for functionality. Here in our implementation, we employ a switch-only rectifier, adjusting power utilization to what can be reasonably harvested with this choice. This topology brings the advantage that it can operate without the need for resonance and, as such, can do so

with low on-chip area consumption. This is also now clarified in Supplementary Section 2.

11. There are many highly efficient switching techniques used in low-frequency piezo power harvesters where the piezo is best modeled by a capacitor and a current source. How does the efficiency of this chosen technique scale to the MHz range? How does it compare with a resonant device in terms of absolute power harvested? I can see that it is qualitatively more efficient from your diagram, but a comparison of volume vs. harvested power would be very interesting.

Response: Switching techniques boost the harvested power when the input impedance of the energy source is dominated by a capacitor and is widely applicable to piezo energy harvesters at low frequency. We extend these techniques to MHz switching frequencies here since these techniques have no fundamental limits in their frequency of operation; higher frequency operation allows for a reduction in the size of the piezoelectric element for overall mote size reduction.

In contrast, resonant power harvesting relies on conjugate matching in which the piezo shows a resistive input impedance at the resonant frequency. Our experiments (see Supplementary Section 1) show that when the piezo is loaded by DI water and an FR4 PCB, mechanical damping reduces the Q of the resonance, showing an overall capacitive input impedance that naturally lead to switch-based harvesting approaches. Resonant power harvesting is still possible with matching inductors.

The overall power conversion efficiency has two main constituents, the efficiency of the acoustic-to-electrical energy conversion and the efficiency of the rectifier itself. The first is reduced by the lack of resonance. One can recover this by using engineered mechanical boundaries (like an air pocket proposed in [10], now [12]), but this introduces extra volume and complexity in packaging. The efficiency of the rectifier itself can be improved by resonating the capacitance of the transducer with an inductor, but this also significantly increases the volume of the device to accommodate this inductor. While resonant power delivery yields to the highest possible efficiency, the goal of the switching techniques applied here is to approach resonant power efficiencies at significant reduced volume requirements. The switch-only approach requires neither off-chip components nor special mechanical boundaries. The switch used in this rectifier is sized with $W/L = 10 \mu\text{m}/600 \text{ nm}$, an almost negligible addition to die area.

We expanded Supplementary Section 2 for clarification. We also include a comparison of achievable power of each method in simulation, with their volume penalty and other concerns explained.

12. The design of the active diode doesn't need to be so prominent in the text since this topology is well-known and was used in ultrasound in Charthad '15.

Response: From our experience, the topology presented in Charthad's paper (now [28]) does not work well in our application, although there is some similarity between this circuit topology and the one employed by us. The key difference is in the terminal " V_{BIAS} ". The Charthad's circuit works for ultrasound waves that have a sufficiently large duty cycle; however, we discovered that no voltage bias (" V_{BIAS} ") can be found that allows this circuit to function correctly with duty cycles for the ultrasound pulses on the scale tens of ppm's as required here. This is primarily because the higher the static current consumption (" I_{BIAS} ") between V_{DC} and ground, the faster the comparator operates. Our topology effectively connects " V_{BIAS} " to " V_{AC2} " in Fig. 6 of [28], allowing dynamic biasing at the cost of reduced efficiency but allowing the circuit to outperform the original topology in case of very sparse ultrasound pulses. This has now been clarified in the "Rectifier Design and Power Regulation" section under "Design Considerations."

13. "and has the potential to scale to sub-0.1-mm³ range implant volumes with piezoelectric integration" Why did you not scale to a smaller volume if this is possible?

Response: Achieving a smaller integration volume requires monolithic integration of piezoelectric element on the CMOS die, as previously demonstrated in Shi, et al. ([24], now [31]). Future work will perform this integration for this mote design. This has now been clarified in the "Post Processing and Packaging" section under "Results".

14. Font sizes in Figure 2 should be increased.

Response: We thank the reviewer for the suggestion. Font size in Figure 2 has been increased for better visibility.

15. Figure 3 is low resolution and hard to read.

Response: We apologize for this. We have reformatted the figure and included a top-level block and timing diagram.

16. In Fig. 3 you show Pz+ going directly into a logic gate. Is there no issue with this?

Response: We thank the reviewer for pointing this out. Here these particular logic gates (inverters) are designed with thick-oxide IO transistors that can tolerate up to 3.3 V. They are further protected by the on-chip ESD protection diodes on all the signal pads and a custom-designed low-leakage power clamp connected between V_{CC} and V_{SS} . This is now mentioned in the "Clock Synchronization and Bi-Directional Data Telemetry" section under "Design Considerations".

17. Fig. 3 would benefit from a higher-level signal timing diagram.

Response: We thank the reviewer for the suggestion. We now incorporate both a block diagram of the entire system and a high-level timing diagram as Figure 3a and Figure 3b, respectively.

18. In Figure 4, what are the dimensions of the PZT and of the chip? Can you please add measurement bars?

Response: We have added scale bars on both the chip's die photo and the photo of the assembled mote. The exact size of each component used in assembling the mote are now clearly noted in the "Post Processing and Packaging" section under "Results".

19. In Figure 5 mechanical index is given at each depth, which is great. Can you also provide intensity values since that is also an important limit for ultrasound?

Response: We thank the reviewer for the suggestion. We now report the maximum intensity value in the "Performance Characterization" section under "Results".

20. In Figure 6 it would be helpful to align the ultrasound image with the high pass image so that it is visually easier to see the location of the motes in the former. Alternatively, the same locations can be labeled in the former.

Response: We thank the reviewer for the suggestion. We have now modified Figure 6 to have the pictures aligned properly with arrows indicating the locations.

21. In Figure 6b there are bright spots surrounding the mote in the high pass image. They are under the circle and therefore hard to see. What are these caused by? Can you move the circle so that these can be seen?

Response: We believe these are caused by rasterization issues when converted to pdf using the submission portal. The original image does not have the bright spots (see picture attached below). We have changed the circles into arrows in our revised submission to resolve this issue.

22. You are using an acoustic absorber for in-vivo measurements. If this is required, how do you anticipate the device can be used in a more realistic environment?

Response: The acoustic absorber is there to reduce the level of ultrasound reflection at the boundary. For the *in vivo* setup, this is placed on the surgery table beneath the mouse's leg. When these implants are used in larger animals, the residual ultrasound energy – the pulses that are not reflected within the field of view, will dissipate within the tissue beyond the field of view. From this perspective, the acoustic absorber actually mimics this situation with larger animals. We have clarified this in the "In vivo Demonstration" section under "Results".

23. The movies are a nice and helpful demonstration. For device 3 in the second movie, the final "0" lights up in a different location. What causes this?

Response: We appreciate the reviewer pointing this out. This artifact resulted from the way the B-mode data was streamed. The hardware of the Verasonics system allows only a finite number of B-mode frames to be stored in its "receive buffer". We believe what happened in this data set is that the first half of the current 16-bit (32 frames) uplink data frame (see Supplementary Table S3) recorded over the top of the previous frame. This frame misalignment can cause artefacts in data recording. We have repeated the experiment, and corrected this error with results from a new *in vivo* experiment. We have replaced Supplementary Movie S2 with this new result. This artifact is no longer present.

Reviewer #2 (Remarks to the Author):

This is solid, novel work but in our opinion, requires additional work to properly substantiate the claims made.

1. A single implant was imaged at a maximum depth of 40 mm in a homogenous medium. In the abstract, the authors claim that an arbitrary number of motes can be imaged within a field of view and that signal-to-noise ratios of 19.1 dB are demonstrated at depths of up to 40 mm in tissue. Additional data is needed to verify this claim in more realistic tissues where multiple motes are implanted.

Response: The authors acknowledge that this statement is misleading and have modified this claim in the abstract. We have successfully demonstrated the identification of more than one mote in ultrasound B-mode imaging, including *in vivo* operation of the mote. Due to the size limitations of the mouse model, we were not able to demonstrate 40 mm of depth *in vivo* yet.

2. We believe that some of this work was previously published in the 2019 IEEE Custom Integrated Circuits Conference. The authors should clarify what is new vs. what has been published.

Response: We thank the reviewer for pointing this out. In our previous work ([15], now [21]) presented in 2019 IEEE Custom Integrated Circuits Conference (CICC), we used the designed integrated circuit to assist postsurgical localization processes. In this paper, we extend what was originally presented in the 2019 CICC by exploring how these implants operate within the context of B-mode imaging, including *in-vivo* applications. This has been clarified now in the Introduction.

Introduction

3. The authors may consider more broadly discussing the state-of-the-art wireless systems based on miniaturized implants. The key advance of this work, namely the wireless power/communication capabilities that enable postsurgical localization of the implants, should be set in the context of the prior body of work in bioelectronics. See, for instance: S. Sonmezoglu, et al. "Monitoring Deep-Tissue Oxygenation with a Millimeter-Scale Ultrasonic Implant," *Nature Biotechnology* 39, 855–864 (2021). Kim, Albert, et al. "An Implantable Ultrasonically-powered Micro-Light-source (μ Light) for photodynamic therapy." *Scientific Reports*, 9.1 (2019).

Response: We thank the reviewer for the suggestion. We have added these additional references. However, the key idea of our work is to demonstrate the mote's ability to be

directly identified in B-mode medical ultrasound, such that the operation of the mote relies on widely adopted hardware platforms in current ultrasound imaging practice and can localize the mote biogeographically while receiving transmitted data; we believe that this work's significance is far higher than just postsurgical localization.

4. The authors do not review existing technologies used for determining the location of miniaturized implantable medical devices inside the body. The authors should compare their localization approach based on ultrasound imaging with the current techniques based on, for example, radio frequency (RF) signals, magnetic fields, or nuclear magnetic resonance.

Response: The key proposal of this work is that we can now integrate the IC-based mote's operation with conventional medical ultrasonography. Medical ultrasound itself has been widely adopted in practice, with its unique advantages explained in our manuscript. Our motes expand what medical ultrasound can image by acting as active contrast agents, providing an "augmented ultrasonography". We do not attempt to argue that our approach is the best for localizing implantable systems, and a comparison with other imaging techniques could be misleading.

Design considerations

5. The power transfer & data communication protocol and the reconstruction algorithms used to form an image are not clear from the treatment in the main manuscript. We recommend adding a timing diagram to demonstrate the power and downlink/uplink data waveforms and simplified block diagrams to explain decoding of the backscattered signals to form an image.

Response: We thank the reviewer for the suggestion. We now incorporate a high-level block diagram, as well as a high-level timing diagram in Figure 3, explaining the operation of the motes in the context of the augmented ultrasonography. The image reconstruction algorithm is based on standard delay-and-sum algorithm commonly used in B-mode imaging (see [27], [35], and [36] for details), with specific parameters used in this work listed in "Methods" – "Image Reconstruction". The backscattered data is then extracted using temporal high-pass filters on the reconstructed B-mode movie.

6. Is the frame rate dependent on the operating depth of the mote due to time-of-flight? For this imaging system, is the operating depth of the mote limited by the fixed frame rate of 50 fps? Please clarify in the text.

Response: 192 pulses with an 81.9- μ s recording time per pulse is used for each frame. This amounts to a maximum frame rate of $1 \text{ s} / (81.9 \mu\text{s} * 192) = 63.6 \text{ fps}$. Each pulse records to a depth determined by twice the time-of-flight equaling the 81.9- μ s recording time. This results in an imaging depth determined by $(81.9 \mu\text{s} * c_s)/2$, where c_s is the speed-of-sound. This amounts to 63.1 mm for $c_s = 1540 \text{ m/s}$. Currently, the operating

depth of the mote is not limited by the frame rate. This is now clarified in both the “Performance Characterization” section under the “Results” and the “Modified Ultrasound Imaging” section under the “Methods”.

7. It is not clear why the authors chose to use two cascaded rectifiers, for example, instead of a single active full-wave rectifier? The authors are expected to characterize the rectifiers and report the related data. Is the voltage clamp used to limit the harvested voltage to 1.2V?

Response: The need to cascade rectifiers comes from the requirement of “switch-only” operation. “Switch-only” operation, as explained in Supplementary Section 2, relies on knowledge of the incoming voltage waveform’s phase information to turn on or off a switch to reduce the dead time during which V_{CC} is not charging from the input source. The required turn-on time for the switch coincides exactly with is the time at which the active diode turns off. If a single-stage rectifier with two active diodes is used, for instance, in the form of Charthad, 2015 (now [25]), then an additional arbitration block is required to generate the switch’s signal from the two independent comparators. We have explained this now in the Supplementary Section 2. While we do not argue that this topology is superior to all other possibilities for ultrasound power harvesting, its ability to enable off-resonance ultrasound power harvesting with minimum area overhead is well-suited to this application.

We have characterized and reported rectifier related the data as Supplementary Section 2, The voltage clamp is designed to limit V_{CC} to 1.2 V.

8. Can the authors provide the measured impedance of the packaged piezoelectric crystal versus frequency in castor oil and indicate the operating frequency and impedance on the figure? Is there a significant impedance mismatch between the piezo crystal and input to the full-wave rectifier at the expected operating voltage? How much can the power harvesting efficiency be improved through impedance matching?

Response: We have reported measured impedance data in Supplementary Section 1 in the cases of PZT submerged in DI water. Water closely matches the acoustic property of human tissue. Castor oil is used in depth related measurements, since it has a similar acoustic impedance to water, while having much higher acoustic attenuation due to its higher viscosity. This mimics the worst-case ultrasound energy attenuation, commonly found in fat layers in the human body.

With water as the mechanical boundary (it is similar for castor oil in terms of acoustic impedance), the impedance of the piezo crystal is dominated by its capacitive component and conjugate matching requires the addition of an external inductor, which will increase the overall size of the mote. Without the inductor, there is a significant impedance mismatch between the piezo crystal and the input to the full-wave rectifier,

especially in the imaginary part. We have now reported simulation data using the measured piezo impedance, and analyzed approaches, including improvements in the mechanical design and conjugate matching, that can potentially boost the power harvesting efficiency in Supplementary Section 2. Here the switch-only rectifier satisfies our power budget requirements and is a suitable approach that allows potential off-resonance operation and opportunities to further minimize the volume of the mote.

9. The authors state that “First, a resonant power delivery requires a high-quality mechanical resonance, which, in turn, demands highly mismatched boundary conditions around the piezo transducer of interest... Second, significant impedance changes typically occur within the approximately 100-kHz bandwidth around the package-defined resonance frequency.” We disagree with this statement. The size of a high-quality factor piezoelectric crystal can be adjusted to achieve impedance matching between the piezo and the input of the rectifier, depending on the application. Furthermore, a high-quality factor piezo achieves significantly higher modulation depth at two extreme piezo terminations (short- and open-circuited piezos) at resonance, improving the uplink data reliability of the system. Can the authors explain why significant impedance changes typically occur in practice?

Response: In the original manuscript, we try to explain two features associated with high-quality factor, resonance-based piezoelectric energy harvesters that make them less suitable for powering motes during B-mode imaging.

First, maintaining a high Q can be problematic. To create such a high-quality-factor resonance, not only does the piezo element itself need to have a high Q, but such Q cannot be lowered by any dissipative mechanical boundaries. In realistic implant scenarios, the mechanical boundary of the mote is very heterogeneous with muscle tissue, fat, and bone all potential in contact. Supplementary Section 1 documents our experiment when the piezo element is loaded by water and the FR4-based PCB, in which we found significant amounts of damping that almost completely eliminates the series resonance. One approach that has been previously explored to maintain the mechanical Q, is the addition of tiny air pockets ([10], now [12]); however, this has impact on packaging complexity, overall size, and the potential risk for air leaking out when implanted.

Second, robust resonance energy harvesting is difficult with commercial medical ultrasound imaging systems. Accurate frequency control is not a common practice in diagnostic ultrasound imaging. In high-Q resonant power harvesting, both the real and imaginary part of the impedance change significantly with small frequency perturbations from resonance (see Figure 3 in [10], now [12] for an example). Similar effects are observed if the inductive band of the piezo is used with a matching capacitor. Without the capability to accurately control the ultrasound's center frequency, it is hard for the imaging system to operate motes precisely at the desired resonant frequency.

Here our approach is not to rely on resonant power harvesting. Although the total power budget of the mote is lowered, we have successfully designed circuits to successfully operation with the available harvested power. We have now tried to make this clearer in the text.

10. Can the authors clarify how to boost the power harvesting without resonance by employing a “switch-only” rectifier in the chip? Furthermore, the authors mention that “Such an approach requires little complexity in packaging...” Does complexity in packaging mean that this approach requires the piezoelectric crystal to be closely integrated with the chip to reduce the electrical parasitics and hence to improve the power harvesting performance?

Response: This is very well explained in Ramadass, 2010 ([23], now [30]) and in the Supplementary Section 2 in our original submission. The complexity in packaging here refers to the addition of air pockets (for high-Q mechanical resonance) or the inclusion of external inductors (for more aggressive resonant power transfer). We have now tried to make this clearer in the text.

11. The authors state “We choose to maximize the modulation depth by shorting the piezoelectric element through MBS (Fig. 3f) when transmitting an uplink bit 0.” The switch (transistor) has an internal resistance when it is turned on to transmit an uplink bit 0. Can the authors provide the switch-on resistance, the electrical load resistance when the switch is off, and the measured backscatter amplitude modulation at the system operating frequency?

Response: We have now included a brief discussion, with measured results, of the switch-on resistance, electrical load resistance, and the measured backscatter modulation depth in Supplementary Section 7. Since the mote now operates under B-mode sonography, the best measure of the uplink data quality is the SNR defined by pixel intensity modulation in the resulting B-mode movie. The measured data modulation depth after image reconstruction is $2/256 = 0.78\%$ in an eight-bit gray-scale movie.

12. The mote powers up when it receives the first ultrasound pulse packet and modulates the backscatter in the second ultrasound pulse packet, which limits the data rate to one-half of the frame rate. Each pulse packet is 1- μ s long. The 1- μ s packet duration seems very short to transmit uplink data from the implant when a sensor is integrated into the mote to monitor biomarkers, such as oxygen and pressure. With respect to the data transfer method, does this system only support analog backscatter amplitude modulation to transmit uplink data as the uplink data appears as a frame-to-frame change in intensity? Would it be possible to use digital backscatter amplitude modulation in this imaging system if multi-bit

digital data from the implant sensor is desired to be transmitted to the external transceiver?

Response: The mote modulates all pulses in a pulse packet for either a 1-bit 1 (M_{NB} open) or a 1-bit “0” (M_{NB} short). The reason why our uplink data rate is limited to one-half of the frame rate is because when M_{NB} is shorted, the pulse packet hitting the mote is absorbed by M_{NB} , preventing the trigger of clock recovery; the clock can only be triggered by the next pulse packet, when M_{NB} has been reset to open asynchronously. Each pulse is 1 μ s long; however, the duration of each pulse packet varies as the relative location between the mote and the imaging probe changes. The width of the pulse packet can vary from less than 0.9 ms (or 9 pulses), when the mote is positioned at the focal depth, to about 3 ms (~30 pulses) when placed in the near field (see Figure 2c and related text for details). The system we designed does not support analog backscatter modulation, but only digital backscatter modulation, i.e., backscatter amplitude shift keying. The frame-to-frame change in intensity is also a digital after image reconstruction (see Supplementary Movies S1 and S2).

The data transmission protocol used between our designed mote and the imaging system is based on ASK. Since in imaging systems, ultrasound energies are sent in the form of short pulses, we find it more robust to implement amplitude shift keying, which is a digital amplitude modulation. In our particular implementation, the average ultrasound pulse width is only four periods long (1 μ s). Since imaging applications require minimum pulse durations for better depth accuracy (z-direction), it is difficult to implement amplitude modulation on such short pulses.

Readout from multiple motes

13. It seems that this imaging system interrogates multiple motes in a time-division multiplexing fashion. In this approach, when many motes are implanted into different locations in a tissue of interest, the frame rate will be limited by the number of motes. What is the maximum number of motes that can be implanted into tissue for the frame rate of 50 fps? Would it be possible to enable simultaneous operation of the motes implanted in different locations of target tissue? Ultrasound reflections from the multiple motes in heterogeneous tissue at similar depths may interfere with each other at the face of the transducer array in a time interval within when the uplink data is received; this may degrade system performance. The operation of the system with multiple motes, at least two motes, at similar depths in explanted animal tissue should be reported.

Response: In our approach, we seek to require minimal changes to how B-mode imaging is performed, while supporting power and data transmission. In this spirit, the only change that has been made over the requirements for imaging is that we are modulating the pulse width used in each frame to support data downlink. In this way, we can support as many motes as will fit into a 50-fps imaging session’s field of view. The newly added Figures 3a and 3b explain this idea using timing waveforms. The frame

rate in this case only limits how deep the recording goes, if the number of scan lines (the lateral, or x, resolution) is fixed. With our current configuration, the field of view is about 38.6 mm (x) by 61.6 mm (z). Since each mote reflects part of the ultrasound energy, conservatively, we can fit up to 20 motes as long as they do not shadow each other from being placed along the same scan line (that is to say, at the same x coordinate, but different z). We further note that we have already demonstrated the operation of motes implanted in different locations simultaneously. Our Supplementary Movie 1 shows exactly this case. Also, during an imaging session, ultrasound energy is focused along a scanline with a finite beam width. That is, within each frame, spatial locations separated apart by at least this beam width in the field of view will see ultrasound energy at different times. In this way, data communications from different motes do not interfere with each other as long as they happen at different time slices in each frame, regardless of whether they are placed at the same depth or not. We have now included Supplementary Section 9 and Supplementary Figure S10, documenting an additional two-mote experiment, where the two motes are placed at a similar depth (34 mm apart from the transducer) with 3.3 mm lateral separation. Our experiment results show no detectable data interference between the two motes in the final B-mode movie.

14. One of our concerns is the long-term reliability of the uplink data transmission between the implant and the external probe, introducing errors in the localization of the implants. How would the system localization accuracy be affected by changes in the angular position of the implanted mote due to external pressure, breathing, or scar formation?

Response: The key idea of this work is that the mote is now visible in conventional B-mode sonography, requiring no targeted focusing but only presence of the mote in the imaging field-of-view to maintain the data transmission. Position of the mote in the field-of-view only requires approximate knowledge of mote location post-implantation. As long as the mote appears in the B-mode image and is able to harvest sufficient energy for operation, it will be functioning and operational, delivering a digital data signature that provides additional contrast with from the surrounding tissue.

15. If multiple motes are closely implanted into different locations in a tissue of interest, ultrasound reflections from the motes may interfere with each other: this may degrade the system localization performance. How close can the motes be positioned in the transverse (x- and y-) and longitudinal (z-) directions in tissue?

Response: In short, in the lateral direction, the minimum spacing between adjacent motes is determined by the beam profile, with typical values around 1 mm before significant interference happen. The motes cannot be stacked on the longitudinal direction.

In the lateral direction, ultrasound beams (or each scanline) occupy a certain width, which varies as the longitudinal distance changes. If the distance between two adjacent motes is smaller than the beam width, then one scanline will carry data from both motes back, causing interference. From the simulated beam profile shown in Figure 2, we estimate a minimum distance of 1 mm between adjacent motes on the transversal direction before significant interference is expected to happen. In practice, we have verified motes' interference-free operation with 3.3 mm of lateral separation at depths of 34 mm in Supplementary Section 9. We have now noted this in the manuscript.

In the longitudinal direction, a mote shadows another mote placed beneath it, as the mote on the top reflects part of the ultrasound energy. In our newly added Supplementary Section 8, we simulated the resulting B-mode image and the maximum pressure on each position in space during the imaging session. Even though the pressure wave can bypass the mote to reach deeper regions, the backscattered signal will again be blocked by the top mote.

16. How would the localization accuracy of the system change with the field-of-view size in heterogeneous tissue?

Response: Here we do not aim at simply localizing the motes, but we integrate their operation with B-mode imaging. In this case, the localization accuracy is the B-mode imager's spatial resolution. Heterogeneous tissue environments affect the resulting image, but as long as the implanted mote is present in the field-of-view with sufficient backscattered power, the location of the motes remains biogeographically accurate in the context of other structures in the B-mode image with uplink data carried within the B-mode image. We have tried to make this point more clearly in this revision.

17. An ultrasound path with higher acoustic attenuation due to absorption and scattering can significantly degrade system data transfer reliability and hence system performance. The localization performance of the system with multiple motes at different locations in explanted tissue with skin and fat content, such as porcine muscle tissue, should be quantified and reported to demonstrate the effect of acoustic attenuation on the system performance.

Response: The goal of this work is to explore motes that can be incorporated into B-mode imaging, with the bi-directional data link embedded into B-mode imaging operation. In Figure 6, we demonstrate multiple motes operating in layered chicken breast cuts submerged in castor oil, with the probe suspending in the castor oil. Chicken breast has a similar acoustic property as that of the human muscle, while castor oil offers a high acoustic attenuation similar to that of fat. This layered structure matches the conditions requested by the reviewer. The in vivo demonstrations show mote operation in even more heterogeneous environments composed of skin, fat, bone, and muscle. We have tried to clarify this further in the manuscript.

18. The modulation depth in an *in vivo* experiment is generally expected to be lower than the modulation depth measured in a homogeneous medium (castor oil), since the ultrasound reflections from internal tissue interfaces likely interfere with the ultrasound reflections from the notes. Does the modulation depth of the backscattered signal depend on the background scattering by the tissues? How does it compare to the castor oil experiment in Fig. 5? What is the maximum modulation depth achieved in castor oil, and what is the minimum detectable modulation depth by this system?

Response: The effective modulation depth of the final received backscatter signal degrades due to background scattering by the tissue. This can be found when comparing our *in vivo* experiment results reported and those in our castor oil experiment. Since this backscatter signal needs to be processed by reconstruction algorithms (details available in the section “Image Reconstruction” in “Methods” and Supplementary Section 6), how this translates to the final image is difficult to quantify. We use the signal-to-noise ratio (SNR) in the final resulting image as a reliable measure of the effective modulation depth. Background scattering effectively adds to the random fluctuation of ultrasound intensity received from the location of interest, which is treated as noise in later processing steps, a point that is now clarified better in the manuscript (“Image Reconstruction” section under “Methods”). In our previously reported *in vivo* experiment, we achieved an SNR of 18.8 dB, which is lower than what we achieve from castor oil experiments, where the environment is homogeneous. We have completed another *in vivo* trial that yields an SNR of 28.5 dB, which is now used in the current revision. At a similar depth in the castor oil experiment, we have achieved SNR of over 30 dB (shown in Figure 5). The degradation of SNR comes from random scattering in the heterogeneous tissue environment, as the reviewer notes. During our experiment, we find the minimum detectable modulation depth from a single transducer is dominated by fluctuations in the received ultrasound due to random scattering. For example, to achieve less than one erroneous bit every 20 frames, a minimum SNR of 10 dB is required. Our data communication protocol (explained in Supplementary Section 5) has parity checks and can be used to detect these errors.

19. It seems from Figs. 5 and 6 that the image quality obtained in the *in vivo* experiment is seen to be lower compared to that obtained in the *in vitro* and *ex vivo* experiments. Can the authors provide the *in vivo* localization accuracy and explain the reasons for degradation in the image quality in the *in vivo* operation?

Response: The resolution of the B-mode images is identical throughout the manuscript. The B-mode images appears “cleaner” in Figure 5 simply because the media is homogeneous and there is no scattering from the complex tissue environment as is present in the *in vivo* image. Such a spatial “noise” (speckle noise) carries the information about the density of random scatterers in the local tissue.

Again, we stress that localization of the motes is not necessary for mote operation; they function in the context of B-mode imaging as shown in Supplementary Movies S1 and S2. Localization accuracy is determined by the resolution of the ultrasound imaging, which in our case is approximately 1 mm in the lateral and transverse directions and 1.5 mm in the axial direction, which is now noted in the manuscript.

20. Can the authors provide estimated depths for the motes operated ex vivo and in vivo in Fig. 6? This can be done, for example, using the time-of-flight information (that is, the time delay between the received backscatter signal and the signal that drove the probe).

Response: The pictures shown in Figure 6 are cropped from the actual B-mode images. In the B-mode reconstruction algorithms, the time-of-flight information is indeed used to create the axial scale, which is the vertical dimension in all B-mode images. The estimated depths of the motes are now clarified in both “Ultrasonography with Phantom” and “In vivo Demonstration” sections under “Results”.

21. The in vivo configuration is based on ultrasound transmission through layered tissues (skin, muscle, bone). The effect of reflections from this layered structure on the system performance should be quantified, as reflections from interfaces at different depths will result in multipath interference that may degrade performance. A possible way is to compare the localization accuracies obtained in the in vitro and in vivo experiments with the motes operated at similar depths.

Response: Here, both our *in vitro* and *in vivo* setups involve layered tissue structures, and quantified performance has been reported in our original submitted manuscript. The localization accuracy is only limited by the resolution of the B-mode imaging process, which is determined by the commercial imaging system and the commercial linear array transducer. Effect from the reflections in layered structures are a part of the resulting B-mode image, independent of the presence of the motes. In our manuscript, we instead use SNR to quantify the performance, since the effect of reflection from layered structures noise to our datalink. The SNR values in all experiments are available in the manuscript.

22. The in vivo experiment was performed with a single mote implanted into the lower hind limb of an anesthetized mouse. The multi-mote in vivo operation should be demonstrated to prove the system’s capability of localizing more than one mote in layered tissues.

Response: We have demonstrated two-mote operation using a standard B-mode imaging probe array. In addition, we have verified mote operation *in vivo*. These experiments are sufficient to demonstrate the key technology presented here; that is,

the integration of an IC-based mote into the conventional B-mode imaging process with distinguishable active data signatures. It is important to note that no localization of the motes is required for operation; the motes must only be present in the field-of-view of the imager.

23. What is the frequency of motion artifacts caused by, for example, breathing that the system can tolerate?

Response: For the uplink data, we window-and-average over 60 frames for data signature detection, effectively removing motion artifacts below 0.8 Hz. This information is available in Section “Image Reconstruction” under “Methods”.

Reviewers' Comments:

Reviewer #1:

None

Reviewer #2:

Remarks to the Author:

The authors have addressed all of my comments. Nice work.

RESPONSE TO REVIEWERS

Reviewer #2 (Remarks to the Author):

The authors have addressed all of my comments. Nice work.

Response: Thank you very much!